# Impact of Seasonal Consumption of Local Tomatoes on the Metabolism and Absorption of (Poly)Phenols in Fischer Rats

**DOI:** 10.3390/nu14102047

**Published:** 2022-05-13

**Authors:** Álvaro Cruz-Carrión, Luca Calani, Ma. Josefina Ruiz de Azua, Pedro Mena, Daniele Del Rio, Anna Arola-Arnal, Manuel Suárez

**Affiliations:** 1Nutrigenomics Research Group, Departament de Bioquímica i Biotecnologia, Universitat Rovira i Virgili, 43007 Tarragona, Spain; ajcruzcarrion@uams.edu (Á.C.-C.); mariajosefina.ruiz@urv.cat (M.J.R.d.A.); manuel.suarez@urv.cat (M.S.); 2Department of Pediatrics, University of Arkansas for Medical Sciences, Little Rock, AR 72205, USA; 3Arkansas Children’s Nutrition Center, USDA-Agricultural Research Service, Little Rock, AR 72202, USA; 4Department of Food and Drugs, University of Parma, 43124 Parma, Italy; luca.calani@unipr.it (L.C.); daniele.delrio@unipr.it (D.D.R.)

**Keywords:** phenolic metabolites, (poly)phenolic compounds, photoperiod, seasonal consumption, *Solanum lycopersicum*

## Abstract

Consuming (poly)phenol-rich fruits and vegetables, including tomato, is associated with health benefits. The health effects of tomato (poly)phenolic compounds have been attributed to their metabolites rather than parent compounds and their bioavailability can be modulated by several factors. This study aimed to evaluate the effect of seasonal consumption of local tomatoes on their (poly)phenol bioavailability. For this, (poly)phenol absorption and metabolism were evaluated by ultra-high-performance liquid chromatography coupled with mass spectrometry and linear ion trap mass spectrometric (uHPLC-MS^n^) after chronic tomato consumption in Fischer rats exposed to three photoperiods mimicking the seasonal daylight schedule. Tomatoes from two locations in Spain (LT, local tomatoes and NLT, non-local tomatoes) were used in this in vivo feeding study. The bioavailability of tomato (poly)phenols depended on the photoperiod to which the rats were exposed, the metabolite concentrations significantly varying between seasons. In-season tomato consumption allowed obtaining the highest concentration of total circulating metabolites. In addition, the origin of the tomato administered generated marked differences in the metabolic profiles, with higher serum concentrations reached upon NLT ingestion. We concluded that in-season tomato consumption led to an increase in (poly)phenol circulation, whereas LT consumption showed lower circulating metabolites than NLT ones. Thus, the origin of the tomato and the seasonal daylight schedule affect the bioavailability of tomato (poly)phenols, which could also affect their bioactivity.

## 1. Introduction

A diet high in fruits and vegetables has long been appreciated as promoting health and reducing cancer risk and cardiovascular disease risk, and is correlated with increased lifespan [1]. Among these diets, the Mediterranean diet could be highlighted, as it contains mostly fruits and vegetables rich in bioactive compounds, which are widely known to contribute to its health benefits [2]. In this regard, tomato (*Solanum lycopersicum*) forms a crucial part of the Mediterranean diet and is recognized worldwide among the most consumed vegetables [3]. Consuming tomatoes regularly has been linked to a lower risk of several types of cancer and cardiovascular disease [2,3]. In one of the last studies, Michaličková et al. [4] concluded that tomato juice may promote healthy lipid metabolism. However, polyphenol enrichment did not provide an additional benefit to cardiovascular health in subjects with stage 1 hypertension. Tomatoes have strong antioxidant properties that are a result of their high content of vitamins C and E, carotenoids, and (poly)phenolic compounds [5]. Specifically, tomatoes contain a number of (poly)phenolic compounds, i.e., flavonoids and phenolic acids, mainly represented by flavanones (naringenin derivatives), flavonols (quercetin and kaempferol derivatives), and cinnamic acid derivatives (chlorogenic, caffeic, and ferulic acids) [6,7]. In this context, the antioxidant and anti-inflammatory activities of tomatoes have been studied. Indeed, a recent study by Abbasi-Parizad et al. [8] investigated the antioxidant power and anti-inflammatory properties of tomato pomace and fermented tomato pomace in a study in which both were found to possess anti-inflammatory properties with aglycone-polyphenol content being the determining factor. Moreover, fermentation was reported to retain the anti-inflammatory properties.

(Poly)phenolic composition in tomatoes varies for several reasons, with genetic, agronomical and environmental factors, growing location, and storage conditions after harvesting being among the most relevant [3,5,9]. In this sense, it has been observed that the total phenolic content of tomatoes grown at 30°~60°N latitude was significantly higher than those at 0°~30°N latitude. Accordingly, the low latitude or high temperature of the geographical origins may cause an alteration in the phenolic accumulation [5]; however, plant secondary metabolites are synthesized in a variety of ways and are dependent on the metabolic capacity of the plant under study, modulated by genetics, environmental factors, and maturity stage [10]. In a further study, Stewart et al. [6] analyzed the occurrence of flavonols in tomatoes cv. Favorita grown in different countries: Spain, South Africa, England, and Scotland. The flavonol content was 21.5, 16.0, 3.4, and 6.6 mg/kg fresh weight, respectively. More recently, Gonzali et al. [11] summarized that anthocyanin phenotype in purple tomatoes is limited to fruit peel and is environmentally dependent, being particularly induced under high light and low temperatures.

Likewise, data on the phenolic characterization of Spanish tomatoes grown in two locations revealed a statistical effect of location in most compounds: tomatoes grown in traditional areas exhibited higher levels of total phenolic content and some phenolic acids, such as caffeic, ferulic, and p-coumaric acids [3]. Moreover, changes in the phenolic content of tomato products during storage have been reported. In fact, the (poly)phenol content decreased during storage of tomato ketchups and juices, possibly because the 3-hydroxy function at the C-ring of flavonoids is not blocked by a sugar moiety [12].

After tomato ingestion, dietary (poly)phenols appear as phase II metabolites at low concentrations in the circulatory system. Some parent compounds or their metabolites pass into the colon, where they are degraded by the local gut microbiota [13]. There is extensive literature showing that these bioactive metabolites are able to reach the target tissues in an effective amount to exert health-promoting effects [14]. Actually, these health benefits are attributed to their metabolites rather than to their natural forms [15] and, hence, a comprehensive understanding of the bioavailability of tomato (poly)phenolic compounds is crucial to comprehend their impacts on health. Several dietary factors may impact (poly)phenol bioavailability. In addition to endogenous factors such as gut microbiota and digestive enzymes, the food matrix can significantly affect the bioavailability of (poly)phenols [16]. In this context, a growing number of studies indicates that gastrointestinal host physiology exhibits circadian variation [17,18]. It has been described previously that the activities of certain enzymes throughout the small intestine of rats exhibited circadian fluctuations [19,20]. The intestinal microbiota also exhibit diurnal oscillations in composition and function [17]. Therefore, alteration of the host physiology conditions the bioavailability and metabolism of phenolic compounds [21]. One study with grapes reported a relationship between metabolites derived from organic and conventional grapes and the photoperiods to which the animals were exposed, highlighting changes due to cultivar [15]. In this context, the study of tomato (poly)phenols, as representative of the Mediterranean diet, is of great interest. Accordingly, this study aimed to determine whether in-season consumption of local Ekstasis tomatoes influences the (poly)phenol bioavailability and metabolism. To our knowledge, this is the first time that the impact of circannual rhythms in the bioavailability of tomato (poly)phenols is evaluated.

## 2. Materials and Methods

### 2.1. Chemicals and Reagents

Salicylic, *p*-coumaric, caffeic, dihydrocaffeic, 3-caffeoylquinic, 4-caffeoylquinic, 5-caffeoylquinic acids, rutin, and quercetin-3-glucuronide were acquired from Sigma-Aldrich (St. Louis, MO, USA). Vitexin was purchased from Extrasynthese (Genay Cedex, France). Caftaric acid was from PhytoLab GmbH & Co. (Vestenbergsgreuth, Germany). Vanillic acid-4-glucoside was from Cayman Chemical (Ann Arbor, MI, USA). Quercetin-3′-sulfate was kindly provided by Professor Alan Crozier (School of Medicine, Dentistry and Nursing, University of Glasgow, Glasgow, UK).

The subsequent standard compounds obtained from Toronto Research Chemicals (Toronto, ON, Canada) that are listed below indicate the commercial names and catalogue number. 3′-Hydroxycinnamic acid-4′-glucuronide (Caffeic Acid 4-β-*D*-Glucuronide, Catalogue N° C080020); 3′-methoxycinnamic acid-4′-sulfate (Ferulic Acid 4-*O*-Sulfate Disodium Salt, Catalogue N° F308920); 3′-methoxycinnamic acid-4′-glucuronide (Ferulic Acid 4-*O*-β-*D*-Glucuronide Disodium Salt, Catalogue N° 308910); 3-(4′-hydroxyphenyl)propanoic acid-3′-glucuronide (Dihydro Caffeic Acid 3-*O*-β-*D*-Glucuronide, Catalogue N° D448705); 3-(4′-hydroxyphenyl)propanoic acid-3′-sulfate (DihydroCaffeic Acid 3-*O*-Sulfate Sodium Salt, Catalogue N° D448710); 3-(3′-methoxyphenyl)propanoic acid-4′-glucuronide (Dihydro Ferulic Acid 4-*O*-β-*D*-Glucuronide, Catalogue N° D448315); 3-(4′-methoxyphenyl)propanoic acid-3′-glucuronide (Dihydro Isoferulic Acid 3-*O*-β-*D*-Glucuronide, Catalogue N° D448940); and 3-(3′-methoxyphenyl)propanoic acid-4′-sulfate (Dihydro Ferulic Acid 4-*O*-Sulfate Sodium Salt, Catalogue N° D448915).

### 2.2. Tomato Samples

Mature, conventional tomatoes (*Solanum lycopersicum* cv. Ekstasis) from two regions of Spain—southeast (Almería, 36°50′17.3″ N 2°27.584′ O; non-local tomato NLT) and northeast (Tarragona, 41°4′29.24″ N 1°3′8.78″ E; local tomato LT)—were acquired from a local producer store in Tarragona. Whole tomatoes were frozen in liquid nitrogen and crushed (Moulinette 1, 2, 3, blanca 700 W, Moulinex, Munich, Germany), followed by freeze-drying for 1 week at −55 °C (Thermo Fisher Scientific, Madrid, Spain). The lyophilized samples were stored in a dry and dark place until use. The (poly)phenolic composition and concentrations of the tomatoes shown in Table 1 and the nutritional characterization (Appendix A Appendix A) were previously determined [22,23].

### 2.3. Study Design

The study and protocol were accepted by the Ethical Committee for Animal Experimentation of the Universitat Rovira i Virgili (reference number 9495). A total of 72 male Fischer 344 (F344) rats from 7 to 8 weeks of age were divided into three groups (*n* = 24) and exposed to simulation of three photoperiods to mimic the seasonal daylight hours (-DH): winter-DH (6 h light/day), spring/autumn-DH (12 h light/day), and summer-DH (18 h light/day), with ad libitum access to water and a standard chow diet (AO4, Panlab, Barcelona, Spain) for 4 weeks. Then, the animals of each photoperiod were assigned to three groups (*n* = 8) and treated by voluntary oral administration for 7 weeks with lyophilized LT and NLT (100 mg/kg body weight (bw)/day, equivalent to 0.34 and 0.37 mg total polyphenols/kg bw/day of LT and NLT, respectively). The control group received 42 mg of a sugar mixture solution (glucose:fructose, 1:1, *m/m*) per kg bw, to provide the equivalent amount of sugar as those given to the tomato-supplemented rats. Here, the treatments are dissolved in water and administered to rats previously trained to eat them. In addition, voluntary ingestion proved to be an effective method for a controlled daily dose administration and avoided increased stress levels that can influence parameters under study [24]. After this period, animals were slaughtered by beheading 1 h after the last administration. Blood samples were collected and centrifuged (2000× *g*, 15 min, 4 °C) to obtain serum and stored at −80 °C until use (Figure 1).

### 2.4. uHPLC-MS^n^ Analyses of Tomato-Derived (Poly)Phenolic Metabolites in Rat Serum

The (poly)phenolic metabolites of LT and NLT in rat serum were extracted as reported previously by Ardid-Ruiz et al. [25]. Samples were directly analyzed by ultra-high performance liquid chromatography (uHPLC) coupled with mass spectrometry (MS), using an Accela uHPLC 1250 apparatus equipped with a linear ion trap MS (LIT-MS) (LTQ XL, Thermo Fisher Scientific Inc., San Jose, CA, USA), fitted with a heated-electrospray ionization (H-ESI-II) probe (Thermo Fisher Scientific Inc., San Jose, CA, USA). Serum metabolite profiling after tomato (poly)phenol intake was evaluated through target full MS2 analyses by monitoring the specific deprotonate molecule (Appendix A Appendix A). The analysis and quantification parameters of serum metabolites after tomato polyphenol intake were the same as those detailed previously by Cruz-Carrión et al. [23].

### 2.5. Statistical Analysis

Results are presented as mean values ± standard error (*n* = 8). For all metabolites found in serum, a two-way analysis of variance (ANOVA) was performed to determine the influence of the factors studied (photoperiod effect (P), treatment effect (T), and photoperiod × treatment interaction effect (P × T)), and then, individually, each intervention was subjected to a one-way ANOVA to establish any differences between means across the three photoperiods, followed by the least significant difference (LSD) post hoc multiple comparison test to identify means that differed. In addition, Student’s *t*-test was used to estimate any differences between LT and NLT metabolites within the same photoperiod. Differences were considered significant at *p* < 0.05. All statistical analyses were performed using the SPSS (SPSS Inc., Chicago, IL, USA) software package.

## 3. Results

### 3.1. Phenolic Metabolites in Serum after Tomato Administration

Contents of serum tomato phenolic metabolites are detailed in Table 2 and Figure 2. Concentration of metabolites detected in the control group was subtracted from each of the treated groups at the respective photoperiods. The serum tomato phenolic profile found in these animals reflected the metabolites of the two last tomato doses administered to rats (i.e., 1 h and 25 h after tomato administration). As a result, a total of seven phenolic metabolites were identified in rat serum. Cinnamic acid derivatives (CADs) and phenylpropanoic acid derivatives (PPADs) detected in this study occurred mainly as sulfate and methyl-sulfate forms. None of the parent (poly)phenols present in tomato were detected, as well as no phase II metabolites of flavonoids.

### 3.2. Effects of Photoperiod in (Poly)Phenolic Metabolites in Serum after Tomato Administration

Remarkably, 3′-methoxycinnamic acid-4′-sulfate, the most abundant metabolite, was significantly affected by the photoperiod (P), as the animals in winter-DH presented lower serum concentrations than those in the other photoperiods, regardless of the LT/NLT treatment received (Table 2). In particular, it was not detected in the group of animals exposed to the winter-DH and LT administration. The CADs did not change among photoperiods or treatments. As for PPADs, the total PPADs concentration was strongly impacted by a photoperiod effect. Individual PPADs values found in the serum of rats exposed to winter-DH were significantly lower than those of rats exposed to summer-DH (photoperiod effect) and 3-(3′-methoxyphenyl)propanoic acid-4′-sulfate was not found in those animals fed with LT and exposed to winter-DH.

In general, the concentration of tomato-derived phenolic metabolites varied according to in-season (autumn-DH) and out-of-season (summer-DH or winter-DH) consumption of local and non-local tomatoes. In fact, different patterns in metabolite concentrations were observed after consumption of each type of tomato (Table 2, Figure 2). The mean of total metabolites showed a tendency (*p* = 0.079, two-way ANOVA) to be less concentrated in the serum of animals exposed to the winter-DH, than those exposed to the autumn/spring-DH.

When taking into account classes of compounds, no statistical variation in CADs concentration was observed between the animal groups exposed to the three seasonal daylight hours after consumption of both LT and NLT (Figure 2). Regarding LT consumption, total PPADs concentration was statistically lower after out-of-season consumption of LT, specifically in winter-DH, than when it was consumed in-season (Figure 2). On the other hand, regarding NLT consumption, total PPADs values were statistically lower in winter-DH compared to the other two photoperiods.

It is important to note that the contribution of each group of metabolites (i.e., CADs or PPADs) to the overall circulating phenolic metabolites also depended on the exposure to seasonal daylight hours, as illustrated in Figure 2, resulting in a more noticeable effect after NLT consumption. Indeed, it appears that the total PPADs amount increased after NLT consumption as animals were exposed to more hours of light, raising 9%, 16%, and 39% in winter-DH, autumn/spring-DH, and summer-DH, respectively. This pattern was obviously opposite to that observed for total CADs concentration, which went from 91% to 84% and 61% as daylight hours increased. On the other hand, after LT consumption, total PPADs concentration (2% of total metabolites) was found to be 38.1 times lower than total CADs concentration in rats exposed to winter-DH. Interestingly, PPADs values increased up to 12% in rats exposed to autumn/spring-DH but decreased in animals exposed to summer-DH.

### 3.3. Effects of the Geographical Origin of Tomatoes in Their (Poly)Phenolic Bioavailability

Whole Ekstasis tomatoes cultivated in two locations of Spain, i.e., LT and NLT, were studied due to their particular (poly)phenolic profile (Table 1). Total phenolic metabolites in serum after NLT administration were higher than those of their LT-administered counterparts, irrespective of photoperiod exposure (Table 2, Figure 2). Particularly, this effect was significant in autumn/spring-DH-stabled and summer-DH-stabled rats, in which the total metabolite concentration was 1.7-fold and 2.2-fold higher after NLT ingestion, respectively. The greatest number of statistical differences between the concentrations of phenolic metabolites post-consumption of LT and NLT was evidenced in animals housed in winter-DH; in fact, all three differences favored NLT intake, as the absence of detectable amounts of 3′-methoxycinnamic acid-4′-sulfate and 3-(3′-methoxyphenyl)propanoic acid-4′-sulfate in serum was observed after LT intake. In the case of animals exposed to autumn/spring-DH, the concentration of 3-(3′-methoxyphenyl)propanoic acid-4′-sulfate in rat serum was significantly higher after NLT administration. Moreover, last but not least, animals housed in summer-DH differed statistically 17.3-fold in 3-(3′-methoxyphenyl)propanoic-4′-sulfate acid concentration after consuming NLT.

## 4. Discussion

A growing body of scientific evidence has reported that in-season fruit consumption produces optimal metabolic responses, while out-of-season consumption may induce erroneous signaling [22,26,27,28]. Among other benefits, it has been emphasized that the consumption of local sweet cherries [27] and local tomatoes [22] may lead to an enhanced antioxidant metabolic response against oxidative stress. Therefore, bioavailability studies of (poly)phenols, as well as the factors that may modulate them, are important to understand the mechanism(s) of their health effect [16]. In this framework, a recent study by our group revealed that important physiological changes in the bioavailability and metabolism of red grape (poly)phenolic compounds may occur under exposure to different photoperiods [15]. To the best of our knowledge, there are no studies evaluating the effect of different light exposure regimens on the bioavailability of tomato (poly)phenolic compounds. The approach chosen, providing 100 mg/kg bw/day of conventionally grown freeze-dried tomatoes (a dose equivalent to consuming approximately 24 g of fresh tomatoes per day for a 70 kg human), may mimic a realistic context where only small amounts of a single fruit or vegetable are consumed every day. This may help to draw physiologically plausible conclusions [29], while addressing this complex and novel topic. In terms of (poly)phenol intake, this dose of LT and NLT resulted in 0.34 and 0.37 mg total (poly)phenols/kg bw/day, respectively. In addition, Fischer 344 rats were selected due to their physiological sensitivity to photoperiods [30].

The only detection of phenolic acid metabolites was consistent with the predominant phenolic acid content in both LT and NLT. No parent (poly)phenols were detected in the rat serum, which is indicative of no cleavage of the sugar moieties and thus of a low probability of absorption of even minute amounts, and in turn only sulfates phase II metabolites were found [13]. In this framework, sulfate and methyl-sulfate conjugates predominated among the metabolites identified in the blood compartment. This suggested a greater action of sulfotransferases and catechol-*O*-methyltransferases than uridine-5′-diphosphate glucuronosyltransferases in conjugated form [13]. Furthermore, CADs, the predominant group of phenolic compounds in both tomatoes, were the main tomato-derived metabolites found in rat serum. This corroborated with the vast literature indicating that cinnamic acid and its derivatives constitute one of the largest and most ubiquitous groups of plant metabolites [31].

The circannual rhythms are an intrinsic timekeeping system that regulates numerous physiological, biochemical, and behavioral processes at intervals of approximately 12 months [32]. By regulating such processes, the circannual rhythm allows organisms to anticipate and adapt to continuously changing environmental conditions [32]. In this regard, seasonal variations in the effects of bioactive compounds have been reported as a function of its time of administration, including its effects on a set of physiological circadian rhythms, i.e., the temporal structure of the organism [33]. Thus, as evidenced by these results, the bioavailability and metabolism of tomato (poly)phenols depends on the photoperiod to which the rats are exposed. In this regard, one of the metabolites strongly impacted by the photoperiod was 3′-methoxycinnamic acid-4′-sulfate (aka ferulic acid-4′-sulfate), which was less concentrated in animals exposed to winter-DH. This is in keeping with the report of Iglesias-Carres et al. [15] showing that ferulic acid bioavailability shows seasonal variations. In fact, it was only detected in one photoperiod after the administration of conventional grapes. Ferulic acid absorption mechanism, due to its low molecular weight and high hydrophilicity, was probably the paracellular pathway reported for other active ingredients with similar characteristics [2]. In this regard, the bioavailability of ferulic acid from tomatoes in humans has been investigated by monitoring the pharmacokinetics of its excretion in relation to intake, with the result that peak time for maximal urinary excretion was approximately 7 h and the recovery of ferulic acid in urine was 11–25% of that ingested [34]. In fact, it has been reported that ferulic acid significantly reduced cadmium-induced hepatic and renal oxidative stress markers and restored antioxidant defense levels in the liver and kidney of male Wistar albino rats [35]. Nevertheless, it should be considered that the caffeic acid derivatives largely exceeded the ferulic acid counterparts in both LT and NLT tomatoes. Thus, it is plausible that the most circulating ferulic acid conjugates are mainly derived from the methylation of absorbed caffeic acid as well as the methylation of dihydrocaffeic acid rather than the metabolism and absorption of native ferulic and dihydroferulic acid.

In addition, total PPADs was also strongly impacted by the seasonal light schedule to which the animals were subjected, thus demonstrating a higher absorption of PPADs in summer-DH animals than in winter-DH animals. Similarly, the findings by Iglesias-Carres et al. [15] also indicated that total PPADs detected in serum post-ingestion of organic and conventional grapes was higher in rats exposed to the L18 (18 h light/day) photoperiod (summer-DH) than in those exposed to the opposite photoperiod (winter-DH).

As for fruit consumption, traditional fruit consumption was marked by the natural harvest season. Today, however, fruits such as tomatoes are marketed throughout the year, giving the option of consuming them out-of-season [14]. However, each fruit has a distinctive composition of (poly)phenols that depends on intrinsic and environmental factors, including growing location and harvest season [3]. It has been suggested that seasonal consumption of (poly)phenol-rich fruits could lead to significant variations in the regulation of mammalian circadian rhythm-dependent physiology and metabolism and that ingestion of fruits out-of-season could trigger an alteration of the characteristic seasonal metabolism [26,36]. Along these lines, the values of total metabolites detected in serum after ingestion of both types of tomatoes was numerically higher when tomatoes were administered in-season (autumn-DH) than out-of-season (winter-DH). This same pattern was observed in the total amount of PPADs. These results agree with those by Iglesias-Carres et al. [15], who found that the total amount of metabolites present in rats exposed to the L6 photoperiod (traditional grape consumption season), regardless of grape variety administration, was higher than when exposed to the L12 (12 h light/day) or L18 photoperiods. When both LT and NLT tomatoes were consumed in-season, higher concentrations of PPADs, including 3-(3′-methoxyphenyl)propanoic acid-4′-sulfate and 3-(4′-hydroxyphenyl)propanoic acid-3′-sulfate, were detected. These metabolites could have been formed due to the action of colonic microbiota. In this regard, the anti-inflammatory properties of microbiota catabolites derived mainly from benzoic acid, phenylacetic acid, and phenylpropanoic acid after consumption of blueberries have been studied by Russell et al. [37]. These findings may contribute to the chemopreventive power of (poly)phenols after degradation in the distal gastrointestinal tract. In addition, chlorogenic acid-derived catabolites, including dihydrocaffeic acid and dihydroferulic acid, tested in combination at physiological concentrations were found to protect human neuronal cells against oxidative stress in an in vitro experimental model [38].

In this study, significant differences were identified in the absorption and metabolism of (poly)phenols in the serum of Fischer 344 rats after chronic administration of tomatoes of different geographical origin of cultivation: LT, local tomato and NLT, non-local tomato. When comparing the metabolites amount after LT and NLT ingestion, the concentration of total metabolites was higher following NLT ingestion in rats exposed to autumn/spring-DH and those exposed to summer-DH. Likewise, all metabolites that varied statistically within the same seasonal daylight schedule were higher after chronic NLT ingestion than their LT equivalent. These significant variations may be associated with the fact that the metabolites found are mostly phenolic acid derivatives and NLT stood out as having the highest amount of this family of phenolic compounds. However, LT was noted for having a higher flavonoid content, possibly because local fruits are harvested close to their consumption date, which reduces their transport and/or storage time. However, the factors modulating variability in occurrence of serum tomato metabolites along with their actions need to be investigated; these factors may be related to other dietary components of this fruit, like non-digestible carbohydrates [39]. Indeed, LT contained a higher content of dietary fiber and, as it is known, it may bind to (poly)phenolic compounds under gastrointestinal conditions, preventing their absorption in the small intestine and promoting the increased passage of these compounds to the colon [16]. Despite the low protein content of tomatoes, LT was distinguished by a higher protein content, which can result in forming (poly)phenol complexes with proteins, leading to alterations in structural, functional, and nutritional characteristics, as well as in the digestibility of both compounds [40]. In this sense, Trombley et al. [41] have shown the bioavailability of (poly)phenols may be affected by the ability of (poly)phenols to interact covalently with proteins. In fact, there is much evidence that increased quantities of protein may restrict the availability and fermentation of (poly)phenols and metabolite formation from the microbiota via complexation [16]. This is consistent with another study that found a reduction in bioavailability of black tea (poly)phenols due to the effect of protein–phenol interactions [42]. Nevertheless, the interaction mechanisms between tomato (poly)phenols and proteins need to be further explored.

## 5. Conclusions

These findings provide useful information on the fate of (poly)phenols in the body following ingestion of tomatoes grown at two different locations. In fact, the bioavailability of tomato-derived (poly)phenols showed a seasonal daylight schedule-dependent effect, with significant variations between groups. In addition, the highest amounts of total metabolites were found in rat serum after in-season consumption of both tomatoes. The amounts of metabolites found differed considerably post-consumption of LT and NLT, presumably due to variations in the phenolic and nutritional composition of each fruit. Further studies focusing on human and microbial metabolites are needed to establish the mechanisms through which tomato-derived (poly)phenols may exert their health-promoting effects.

## Figures and Tables

**Figure 1 nutrients-14-02047-f001:**
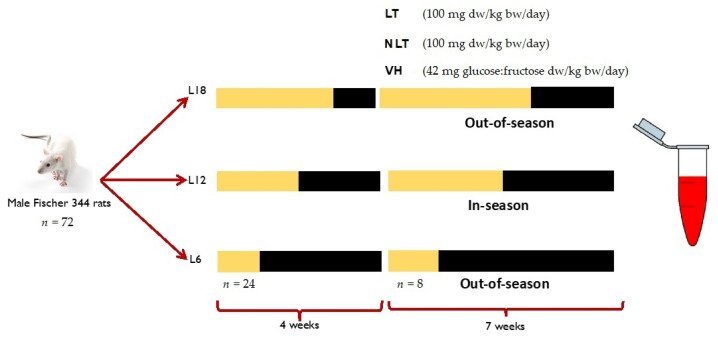
Experimental design of this study. Abbreviations: L18: 18 h light/day; L12: 12 h light/day; L6: 6 h light/day; LT: local tomatoes; NLT: non-local tomatoes; VH: vehicle; dw: dry weight; bw: body weight. The yellow indicates light hours per day of each photoperiod and the black corresponds to darkness hours per day.

**Figure 2 nutrients-14-02047-f002:**
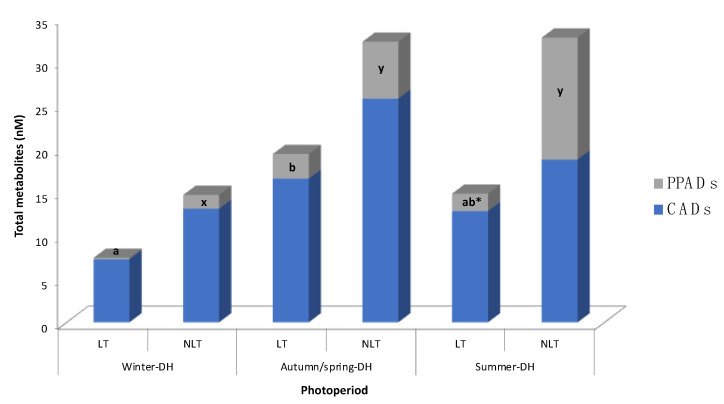
Distribution of (poly)phenolic metabolites in rat serum after 100 mg/kg bw/day administration of local (LT) or non-local (NLT) Ekstasis tomatoes and exposed to seasonal daylight schedules. * indicates statistical difference of PPADs concentrations (*p* < 0.05) between LT and NLT ingestion within each photoperiod estimated by Student’s *t*-test. Values with different letters (a, b; for LT consumption) and (x, y; for NLT consumption) indicate different PPADs concentrations (*p* < 0.05) between the photoperiods, estimated by one-way ANOVA. CADs concentration had no statistical difference by one-way ANOVA or Student’s *t*-test. Abbreviations: CADs, cinnamic acid derivatives; DH, daylight hours; PPADs, phenylpropanoic acid derivatives.

**Table 1 nutrients-14-02047-t001:** (Poly)phenolic composition of local (LT) and non-local (NLT) tomatoes. Data expressed as mean values in µg/g dry weight ± SD (*n* = 3).

Compound	LT	NLT
**Flavonoids**		
Kaempferol-*O*-rutinoside ^a^	3.51 ± 0.48	9.27 ± 1.25 *
Kaempferol-*O*-rutinoside-*O*-pentoside ^a^	4.81 ± 0.58	11.22 ± 2.02 *
Luteolin-*O*-hexoside-C-hexoside ^a^	34.76 ± 3.6	21.2 ± 1.02 *
Naringenin	1.64 ± 0.27	2.18 ± 0.47
Naringenin chalcone ^b^	1.12 ± 0.20	1.43 ± 0.11
Phloretin 3′,5′- di-*C*-β-glucopyranoside ^c^	107.97 ± 11.5	51.19 ± 4.19 *
Quercetin-*O*-dihexoside ^a^	5.87 ± 0.76	5.20 ± 0.34
QHRP-*O*-hexoside ^a^	1.79 ± 0.43	1.24 ± 0.56
QHRP-coumaric acid ^a^	21.16 ± 1.13	10.13 ± 1.57 *
QHRP-ferulic acid ^a^	10.98 ± 0.4	6.86 ± 0.12 *
QHRP-sinapic acid ^a^	4.24 ± 0.16	3.66 ± 0.37
QHRP-syringic acid ^a^	6.55 ± 0.81	8.28 ± 1.54
Quercetin-*O*-rutinoside-*O*-hexoside ^a^	2.52 ± 0.27	0.87 ± 0.13 *
Quercetin-*O*-rutinoside-*O*-pentoside ^a^	81.03 ± 2.80	64.25 ± 5.82 *
Rutin	111.33 ± 1.11	86.38 ± 11.77 *
Total, flavonoids	399.32 ± 16.15	283.37 ± 30.38 *
**Caffeic and dihydrocaffeic acid derivatives**		
Caffeic acid derivative I ^d^	33.24 ± 1.12	65.55 ± 17.50 *
Caffeic acid derivative II ^d^	28.10 ± 2.15	14.81 ± 3.34 *
Caffeic acid derivative III ^d^	13.95 ± 0.68	24.97 ± 6.96
Caffeic acid derivative IV ^d^	4.19 ± 0.10	5.37 ± 0.47 *
Caffeic acid derivative V ^d^	4.79 ± 0.41	4.91 ± 0.98
Caffeoylmalic acid ^d^	62.93 ± 3.55	66.80 ± 9.10
Dihydrocaffeic acid derivative ^f^	38.14 ± 2.12	32.09 ± 5.61
Total, caffeic and dihydrocaffeic acid derivatives	185.34 ± 10.14	214.49 ± 43.97
**Free phenolic acids**		
Caffeic acid	27.30 ± 3.03	40.92 ± 1.25 *
Dihydrocaffeic acid	n.d.	8.72 ± 2.73 *
*p*-Coumaric acid	15.57 ± 1.14	16.19 ± 2.02
Salicylic acid	29.04 ± 3.22	54.39 ± 12.34 *
Total, free phenolic acids	71.90 ± 7.39	120.22 ± 18.34 *
**Hydroxybenzoic acid derivatives**		
Dihydroxybenzoic acid-*O*-pentoside ^g^	24.44 ± 1.46	18.30 ± 1.64 *
Hydroxybenzoic acid-*O*-hexoside ^g^	40.36 ± 1.28	33.59 ± 3.08 *
Syringic acid-*O*-hexoside ^g^	37.53 ± 0.93	37.53 ± 5.59
Total, hydroxybenzoic acid derivatives	102.34 ± 3.66	89.42 ± 10.30
**Hydroxycinnamic acid derivatives**		
Caffeic acid-*O*-hexoside I ^g^	169.42 ± 5.07	235.81 ± 8.06 *
Caffeic acid-*O*-hexoside II ^g^	67.52 ± 4.03	60.86 ± 5.82
Caffeic acid-*O*-hexoside III ^g^	194.60 ± 4.23	159.57 ± 19.72 *
Coumaric acid derivative	14.08 ± 1.85	15.00 ± 1.15
Coumaric acid-*O*-hexoside I ^g^	87.86 ± 2.09	132.38 ± 1.72 *
Coumaric acid-*O*-hexoside II and III ^g^	161.79 ± 7.83	113.07 ± 0.57 *
Dicaffeoyl-*O*-hexoside ^g^	72.09 ± 1.97	109.38 ± 14.05 *
Ferulic acid-*O*-hexoside ^g^	72.85 ± 2.58	24.30 ± 2.35 *
Sinapic acid-*O*-hexoside ^g^	27.05 ± 2.16	31.30 ± 3.01
Total, hydroxycinnamic acid derivatives	867.26 ± 31.81	881.67 ± 56.44
**Hydroxycinnamoylquinic acids**		
3-*O*-Caffeoylquinic acid	30.09 ± 4.38	44.24 ± 4.39 *
4-*O*-Caffeoylquinic acid	223.02 ± 7.43	247.29 ± 7.59 *
5-*O*-Caffeoylquinic acid	200.14 ± 33.63	280.73 ± 9.04
Caffeoylquinic acid-*O*-hexoside I ^j^	39.64 ± 2.68	37.24 ± 5.62
Caffeoylquinic acid-*O*-hexoside II ^j^	52.89 ± 5.94	49.50 ± 5.53
Coumaroylquinic acid ^j^	93.38 ± 9.33	96.80 ± 4.70
Dicaffeoylquinic acid I ^i^	88.27 ± 3.34	103.29 ± 8.83
Dicaffeoylquinic acid II ^h^	39.97 ± 1.91	68.08 ± 1.60 *
Dicaffeoylquinic acid III ^i^	75.77 ± 2.30	155.79 ± 0.01 *
Dicaffeoylquinic acid-*O*-hexoside ^h^	44.75 ± 2.45	37.09 ± 1.56 *
Feruloylquinic acid ^j^	31.76 ± 1.34	31.81 ± 2.03
Tricaffeoylquinic acid ^h^	177.11 ± 3.55	390.57 ± 61.65 *
Tricaffeoylquinic acid-*O*-hexoside ^i^	53.05 ± 4.62	17.93 ± 0.69 *
Total, hydroxycinnamoylquinic acids	1149.83 ± 82.90	1560.35 ± 113.23 *
**Phenylpropanoic acid-glycosides**		
Dihydrocaffeic acid-*O*-hexoside I ^g^	63.44 ± 3.84	74.47 ± 4.54
Dihydrocaffeic acid-*O*-hexoside II ^g^	144.74 ± 4.37	160.71 ± 14.94
Dihydrocaffeoyl-caffeoyl-*O*-hexoside ^g^	79.63 ± 4.81	95.00 ± 18.17
Dihydroferulic acid-*O*-hexoside ^g^	152.76 ± 9.93	94.48 ± 3.04 *
Hydroxyphenylpropionic acid-*O*-hexoside ^g^	145.74 ± 6.55	171.36 ± 0.72 *
Total, phenylpropanoic acid-glycosides	586.30 ± 29.49	596.02 ± 41.42
Total, (Poly)phenolic compounds	3362.30 ±189.92	3745.54 ± 315.02

* indicates significant differences (*p* < 0.05) between LT and NLT by Student’s *t*-test. Abbreviations: n.d., not detected; QHRP, Quercetin-*O*-hexoside-*O*-rhamnoside-*O*-pentoside; SD, standard deviation. ^a^ Tentatively quantified using the calibration curve of rutin. ^b^ Tentatively quantified using the calibration curve of naringenin. ^c^ Tentatively quantified using the calibration curve of vitexin. ^d^ Tentatively quantified using the calibration curve of caffeic acid. ^f^ Tentatively quantified using the calibration curve of dihydrocaffeic acid. ^g^ Tentatively quantified using the calibration curve of vanillic acid-glucoside. ^h^ Tentatively quantified using the calibration curve of 3-*O*-caffeoylquinic acid. ^i^ Tentatively quantified using the calibration curve of 4-*O*-caffeoylquinic acid. ^j^ Tentatively quantified using the calibration curve of 5-*O*-caffeoylquinic acid.

**Table 2 nutrients-14-02047-t002:** Tomato-derived (poly)phenolic metabolites in serum of rats exposed to three seasonal daylight schedules after ingestion of 100 mg/kg bw/day local (LT) and non-local (NLT) Ekstasis tomatoes. Results are expressed as the mean values in nM ± SEM (*n* = 8).

Metabolite		Serum Concentration (nM)						
		Winter-DH		Spring/Autumn-DH		Summer-DH		2wA
		LT	NLT	LT	NLT	LT	NLT	
CADs	3′-methoxycinnamic acid-4′-sulfate	n.d. ^a,^*	3.8 ± 3.8 ^x^	11.3 ± 5.6 ^b^	18.5 ± 9.1 ^x^	8.8 ± 4.9 ^b^	13.1 ± 6.1 ^x^	*p*
	4′-hydroxycinnamic acid-3′-glucuronide	n.q.	n.q.	n.q.	n.q.	n.q.	n.q.	n.s.
	4′-methoxycinnamic acid-3′-sulfate	n.d.	n.q.	n.q.	n.q.	n.q.	n.q.	n.s.
	Hydroxycinnamic acid sulfate I	n.q.	n.q.	n.q.	n.q.	n.q.	n.q.	n.s.
	Hydroxycinnamic acid sulfate II	7.2 ± 3.7 ^a^	9.2 ± 2.0 ^x^	5.2 ± 1.3 ^a^	7.2 ± 1.3 ^x^	4.0 ± 1.3 ^a^	5.6 ± 2.1 ^x^	n.s.
PPADs	3-(3′-methoxyphenyl)propanoic acid-4′-sulfate	n.d. ^a,^*	1.4 ± 0.9 ^x^	0.8 ± 0.3 ^b^*	2.6 ± 1.2 ^x,y^	0.5 ± 0.5 ^b,^*	9.4 ± 3.7 ^y^	P, T
	3-(4′-hydroxyphenyl)propanoic acid-3′-sulfate	n.q. ^a^	n.q. ^x^	2.0 ± 0.8 ^b^	3.9 ± 1.3 ^y^	1.5 ± 0.8 ^b^	4.7 ± 2.0 ^y^	P
Total metabolites		7.4 ± 3.7	14.7 ± 6.1	19.3 ± 6.8	32.3 ± 10.8	14.8 ± 6.6	32.7 ± 11.9	n.s.

* denotes significant differences (*p* < 0.05) between treatments (LT and NLT) within each photoperiod exposure (winter-DH, spring/autumn-DH, and summer-DH), estimated by Student’s *t*-test. Values with different letters (^a^, ^b^; for LT consumption) and (^x^, ^y^; for NLT consumption) indicate statistical differences (*p* < 0.05) of the same treatment among the photoperiods, estimated by one-way ANOVA. Two-way ANOVA was used to evaluate the differences between the groups, P, photoperiod effect; T, treatment effect; P × T, photoperiod × treatment interaction effect. Abbreviations: SEM: standard error of the mean; CADs: cinnamic acid derivatives; winter-DH: winter daylight hours, 6 h light/day; spring/autumn-DH: spring/autumn daylight hours, 12 h light/day; summer-DH: summer daylight hours, 18 h light/day; n.d.: not detected; n.s.: no significant; n.q.: not quantified; PPADs: phenylpropanoic acid derivatives, 2wA: Two-way ANOVA.

## Data Availability

Not applicable.

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
