# Peer review of "Impact of Seasonal Consumption of Local Tomatoes on the Metabolism and Absorption of (Poly)Phenols in Fischer Rats"

_nutrients, 2022, doi:10.3390/nu14102047_

Round 1
Reviewer 1 Report
The manuscript entitled “Impact of Seasonal Consumption of Local Tomatoes on the Me-2 tabolism and Absorption of (Poly)Phenols in Fischer Rats” by Álvaro Cruz-Carrión et al. aimed to determine whether in-season consumption of local Ekstasis tomatoes influences the (poly)phenol bioavailability and metabolism in an in vivo model. The topic is interesting and can contribute to answering many questions, however, it requires a more detailed analysis of the information so that the described methodology really responds to the proposed objectives. I add some important comments:
Line 52. “or high temperature of the geographical origins may lead to a lower phenolic accumulation [4] “Authors should be careful with this type of statement, since it is well known that stress conditions, such as high (extreme) temperatures, activate a response in plants by inducing the production and accumulation of different secondary metabolites, among they, the polyphenols
Line 109 -110 “Whole tomatoes were frozen in liquid nitrogen and 109 grounded, then freeze-dried for one week at -55 °C “Please indicate if you monitor the particle size obtained at the end of the grinding process, or at least the type of equipment used for it.
112-113 “The (poly)phenolic composition and concentrations of the toma-112 toes showed in Table 1 and the nutritional characterization (Supplementary Table 1) were 113 previously determined [18,19]” Given that the results of Table 1 and Supplementary Table 1 are not part of the results obtained in this work, I consider that they should not be included in materials and methods, they should go in the introduction and specifically indicate what you want to highlight from that information.
Table 1. In the Table 1, letters are indicated in each of the compounds, but their meaning is not indicated.
Line 127. “by voluntary oral administration for 7 weeks” If the intake was voluntary, as the authors ensure that the amount of product ingested by each animal was equivalent in order to be able to make comparisons between groups.? Was the lyophilisate administered as such or were pellets made? How was the dose to be administered to the experimental groups defined?
Line 129 “ control group received 42 mg of a sugar mixture solution 129 (glucose:fructose, 1:1) per kg bw” Why was it defined that the best control diet option would be a sugar mixture solution (glucose:fructose, 1:1) per kg bw, and not a commercial control diet, such as AIN-93? What objective was raised with this diet based on sugars?
Table S2 Why was a quercetin derivative used to quantify naringenin, if the maximum length of absorption does not correspond?
Table 2. I consider that the statistics in Table 2 can be improved, especially the way of expressing it to make it easier to understand. The comparison "(x, y, z; 170 for NLT consumption) indicate statistical differences (p < 0.05) between the photoperiods" is not clear in the table. Is this comparison between a group of compounds? or between all the metabolites detected for that photoperiod? I consider that it could be of value to also analyze the total of compounds of each treatment and by photoperiod
Figure 2 presents the same information as Table 1, so information is being duplicated, I recommend delete it.
What were the results of the control group used for?
I consider that the study presents serious limitations in terms of the proposed objective and the experimental execution to fulfill it. If the objective was to evaluate how the different photoperiods affect the bioavailability of phenolic compounds present in the tomato, the most appropriate thing was to evaluate the bioavailability of the same tomatoes but without applying a photoperiod.
Author Response
Manuscript ID: nutrients-1711945
Title of manuscript: Impact of Seasonal Consumption of Local Tomatoes on the Metabolism and Absorption of (Poly)Phenols in Fischer Rats
Dear reviewer, please find point-by-point response to each of your comments with the description of the changes made in the manuscript.
Thank you very much for your attention reconsidering our manuscript for its publication in Nutrients.
Yours sincerely,
Dr. Anna Arola Arnal
Corresponding author

Reviewer 2 Report
The presented work concerns the influence of seasonal tomato consumption on the metabolism of polyphenols present in tomatoes. The study was carefully designed and conducted. The methodology does not raise any objections.
The issue itself is also very interesting and innovative, which should be emphasized.
However, I have some comments that I believe may improve the quality of the manuscript.
1. Introduction
- To be expanded. There is no reference to the content of polyphenols in tomatoes growing at significantly different latitudes (eg Eastern Europe?). It is also worth commenting on other varieties (e.g. purple tomatoes https://www.mdpi.com/2076-3921/9/10/1017)
- I would like to ask for a comment, what is the reason for the increase in the number of polyphenols during the storage of processed tomatoes (lines 58-59)
- I would like to present and comment on the change in the content/bioavailability of polyphenols such as lycopene during the thermal processing of tomatoes or fermentation (https://www.mdpi.com/2076-3921/9/2/179)
- In the introduction, I also miss the reference to research on humans. I understand that rats were used in this experiment, but the introduction is a place to highlight current human research (e.g. https://link.springer.com/article/10.1007/s11130-019-0714-5)
M&M
Are the results in table 1 given per 1 g of dry matter?
Results:
Please comment on table 2. For PPAD's in winter, the results are 0.2 +/- 0.2. The result is equal to the error, it is worth commenting on.
In my opinion, it is worth editing the text by a professional English editor before publication.
Author Response

(The authors gave the same response as above.)

Reviewer 3 Report
The manuscript “Impact of Seasonal Consumption of Local Tomatoes on the Metabolism and Absorption of (Poly)Phenols in Fischer Rats” elucidates tomato and the seasonal daylight schedule affect the bioavailability of tomato (poly)phenols. This study has important reference value for the development of safe metal concentration standards in medicinal plants. However some information needs improve.
- I didn't find what instruments and methods were used to determine phenolic compounds in tomatoes? What is the method of determining the nutrient content in tomatoes that has not been found? The method for measuring metabolites in rat serum is not detailed enough. For example, mobile phase and ion source information, etc.
- Please supplement the determination of the total phenol content and total flavonoid content of these two tomatoes.
- The experimental design is innovative, but I think there are still shortcomings. The total phenolic compounds in NLT were more than LT, but no significant differences were observed. Therefore, I cannot conclude whether there is a correlation between the high content of total phenolic compounds and the concentration of total circulating metabolites? If the experimental design adopts tomatoes from three different sources, and there is a significant difference in the content of phenolic substances (low-dose tomato, medium-dose tomato and high-dose tomato), I think this may explain more problems.
- In animal experiments, serum is collected after 1 h of administration.,However, in 3.1 it is 1 hour and 25 hours. I guess this is due to the difference in lighting conditions? It is recommended to add a schematic diagram of serum collection.
- Why are metabolites in feces not measured?
- The origin of tomato and the seasonal daylight schedule affect the bioavailability of tomato (poly)phenols, which could also affect their bioactivity. So, how do we choose to eat tomatoes?
Author Response

(The authors gave the same response as above.)

Reviewer 4 Report
As is mentioned in the study, this is the first time that the impact of circannual rhythms in the bioavailability of tomato (poly)phenols is evaluated.
This is a preliminary study, which requires further studies, to establish the mechanisms "through which tomato-derived (poly) phenols may exert their health-promoting effects", as mentioned in the conclusions.
The study should have small corrections, explanations:
-in Table 1, in the footer I did not find noted what are the meaning of the letters a-j.
-the bibliography requires writing according to the requirements of the journal
Author Response

(The authors gave the same response as above.)

Round 2
Reviewer 2 Report
The Authors have improved the manuscript. In the present version it has much better general scientific soundness and in my opinion, may be accepted for publication in Nutrients.
Reviewer 3 Report
All questions have been replied.